# Interaction between the place of residence and wealth on the risk of overweight and obesity in Bangladeshi women

Kabir Ahmad[1,2,3¤]*, Taslima Khanam[1], Syed Afroz Keramat[2,3,4], Md. Irteja Islam[2,3,5], Enamul Kabir[2,6], Rasheda Khanam[2,3]

**1** Research Unit, Purple Informatics, Dhaka, Bangladesh, **2** School of Commerce, Faculty of Business, Education, Law and Arts, University of Southern Queensland, Toowoomba, Queensland, Australia, **3** Centre for Health Research, University of Southern Queensland, Toowoomba, Queensland, Australia, **4** Economics Discipline, Social Science School, Khulna University, Khulna, Bangladesh, **5** Maternal and Child Health Division, International Centre for Diarrhoeal Disease, Bangladesh (icddr,b), Dhaka, Bangladesh, **6** School of Sciences, Faculty of Health, Engineering and Sciences, University of Southern Queensland, Toowoomba, Queensland, Australia

¤ Current address: School of Commerce, Faculty of Business, Education, Law and Arts, University of Southern Queensland, Toowoomba, Australia
* kabir.ahmad@usq.edu.au, purple.informatics@gmail.com

**Data Availability Statement:** The datasets of Bangladesh Demographic Health Survey (BDHS) are available at the web archive of DHS program. This is an open sources data-set, which is available

## Abstract

### Background

The prevalence of overweight and obesity in women has increased significantly over the last few decades in Bangladesh, a rapidly urbanising developing country. However, little is known regarding the association between the interaction of the place of residence and household wealth with overweight and obesity, particularly in women from developing countries.

### Objective

The objective of this study is to find the association between the interaction of the place of residence and wealth with overweight and obesity among Bangladeshi women.

### Methods

This study utilised data from the four Bangladesh Demographic Health Surveys conducted in 2004, 2007, 2011 and 2014 with a total of 54337 women aged 15–49 years. Multivariate logistic regression was used for the analyses.

### Results

The prevalence of overweight and obesity among women aged 15–49 years in Bangladesh has considerably increased from 9.96% in 2004 to 24.43% in 2014. The interaction between wealth and place of residence has been found to be associated with obesity. Urban wealthy and richest women were 4.23 (OR: 4.23, 95% CI: 1.25–14.34) and 5.99 (OR: 5.99, 95% CI: 1.91–18.74) times more likely to be obese compared to their rural counterparts in the period

on request at (https://dhsprogram.com/data/available-datasets.cfm). In order to get an access to DHS datasets, authors needs to do followings: 1. Create an account in the DHS program website; 2. Login and agree to the term of data use; 3. Fill up the data application form with required information.

**Funding:** Please note that, this research did not receive any specific grant from any funding agencies in public, commercial or not-for-profit sectors. Two authors (KA and TK) have affiliation from commercial organization, Purple Informatics. The funder provided support in the form of salaries for TK, and consultancy fee for KA, but not for this paper work; and the funder did not have any additional role in the study design, data collection and analysis, decision to publish or preparation of the manuscript for this study. The specific roles of these authors are articulated in the 'author contributions' section.

**Competing interests:** TK is an employee and KA is a consultant of the commercial affiliation, Purple Informatics. These do not alter our adherence to PLOS ONE policies on sharing data and materials. Other authors do not have any competing interests.

2004. Urban richest were 2.94 times (OR: 2.94, 95% CI: 1.20–7.24) more likely to be obese against their rural counterparts for the survey year 2014.

## Conclusions

The place of residence is not associated with obesity, but its interaction with wealth is significant.

## Introduction

Overweight and obesity continue to be serious public health concerns [1–3]. Globally, the prevalence of overweight and obesity is increasing rapidly, with about two billion overweight or obese people across the world in 2016. It has been estimated that 39% and 13% of the world's adult population (aged 18 years and above) were overweight and obese in 2016, respectively [4]. Overweight and obesity are considered as a leading risk factor for the global burden of obesity-related morbidity and mortality [5, 6], with an estimated 3.4 million global deaths in 2010 were attributed to overweight or obesity [7, 8]. Although increasing trends in overweight and obesity have been widely recognised in developed countries [9, 10], these are now concerns in many developing countries due to their high incidence rates and increasing mortality rates in the past two decades [1, 2]. Several studies reported that overweight and obesity are strongly linked to non-communicable diseases such as diabetes mellitus, hypertension, cardiovascular disorders, cancers and musculoskeletal disorders [7, 11], which are more common in women than men [12]. For instance, some forms of cancer such as breast cancer, ovarian cancer and uterine cancer, and adverse birth outcomes such as preterm birth, low birth weight and stillbirth have been found to be significantly associated with overweight and obese women [13, 14]. Bangladesh, a low-income and the most densely populated country in the world, has experienced alarming rates of overweight/obesity due to current demographic and nutritional transition, rapid urbanisation, and modifications in dietary and lifestyle patterns [3, 5, 8]. Recent studies have reported that the prevalence of overweight and obesity in Bangladesh has increased from 9% to 39% between 1999 and 2014 [3, 15]. Evidence also suggests that the burden of overweight/obesity is found to be significantly higher among women than men in Bangladesh [10, 12], with negative long term consequences more in women than men [16, 17]. For instance, a study reported that the prevalence of overweight and obesity among urban and rural women in Bangladesh had been increased by 17.5% and 10.4% in the years between 1999 and 2011 [17, 18]. Moreover, another recent study conducted by Kamal et al. [14] in 2015 found that the prevalence of overweight among married women in Bangladesh is 29.2%, which is alarming.

Worldwide, many studies have investigated the correlates of overweight and obesity among children and adults; however, little is known concerning women living in low-income countries [5, 17, 18]. The majority of previous studies conducted in developing countries, including Bangladesh, have focused on the prevalence of overweight and obesity in women, and tested to what extent a single socio-economic, reproductive or lifestyle factor influences women to be overweight or obese [1, 2, 5, 14]. At the same time, only a few studies have examined the effect of an interaction term between two socio-demographic factors on overweight and obesity in women. For example, a study involving multiple middle-income countries tested the obesogenic effect of the co-existence of education and household wealth in women [11]. Further, even though urbanisation and high socio-economic status are critical determinants of

overweight and obesity in adults [7, 18, 19], to the best of our knowledge, none have examined the effect of an interaction term between household wealth and the place of residence (urban/rural) on overweight and obesity among women in low-income countries like Bangladesh [3, 12, 15]. This lack of examination highlights the importance of investigating the effect of various socio-economic risk factors, including the interaction term between the place of residence and household wealth on overweight and obesity among women in particular [11, 19].

This study investigates whether the interaction between the place of residence and household wealth is associated with overweight and obesity among Bangladeshi women using the nationally-representative sample from the Bangladesh Demographic and Health Survey (BDHS) datasets.

## Materials and methods

### Data source and sample selection

This study analysed publicly available data from the Demographic and Health Surveys of Bangladesh conducted in 2004, 2007, 2011 and 2014, which are nationally representative household-based cross-sectional surveys of non-institutional Bangladeshi people. These surveys are the source of the significant cradle of data reflecting the status of Bangladeshi women on many demographic and health issues, including obesity, collecting through standard model questionnaires using standardised measurement tools to ensure standardisation and comparability across time and geographic locations. These credential surveys were conducted under the supervision of the Ministry of Health and Family Welfare and the National Institute of Population Research (NIPORT), funded by the United States Agency for International Development.

This study excluded participants below 15 years of age, participants with missing values for measured height and weight, and women who were pregnant at the time of the survey. Observations with missing values for the confounding variables considered in this study were also excluded. After applying the exclusion criteria, the total study population from all the four periods of surveys was 54337 women aged 15–49 years. A two-stage stratified sample was used to select the subjects–at the first stage, enumeration areas were selected with a probability proportional to population size, and in the second stage, a systematic sampling technique was used to select households from the enumeration area to provide statistically reliable estimates of key demographic and health variables for the country. Additionally, sampling weight was applied to ensure the actual representation of the survey results at the national and domain levels. Details of the sampling methods are available from the final report of the surveys [20–23].

### Measures

**Outcome and exposure variables.** Body Mass Index (BMI) is the primary outcome variable of the study, which is calculated as weight in kilograms divided by height in squared metres. In this analysis, WHO suggested BMI classification for the world population was used for defining overweight (BMI 25.00–29.99 kg/m$^2$), and Obesity (BMI $\geq$30 kg/m$^2$). This study merged underweight (BMI<18.5 kg/m$^2$) and healthy weight (BMI 18.5 to <25 kg/m$^2$) to form a new category of BMI <25 kg/m$^2$.

In this study, explanatory variables were selected based on the existing literature on the socio-demographic and behavioural factors associated with obesity [8, 12, 15, 19, 24]. Age, education status or wealth index of the respondents relates to different levels of physiological and social status [12, 15, 17], while watching television indicates the sedentary behaviour of more sitting time [8]. Besides considering the literature, the significance of the chi-square tests between the primarily selected independent variables and the outcome variable (BMI category) from the bivariate analyses of BDHS data were considered to validate the rationale of choosing

these variables. These explanatory variables were: age (15–24, 25–34, 35–49), divisions (Barisal, Chittagong, Dhaka, Khulna, Rajshahi, Sylhet), place of residence (rural or urban), education status (no education, primary, secondary, Higher), marital status (married, widowed, divorced/not living together), parity (0, 1, 2, 3, $> = 4$), watching television (not at all, less than once a week, at least once a week), wealth index (poorest, poorer, middle, richer, richest), currently working (no, yes) and contraceptive use (not using, hormonal, non-hormonal and traditional methods).

**Statistical analysis.** Descriptive statistics on sampling characteristics and prevalence of overweight/obesity were computed for all the four survey years. Stata 14 software was used for all the statistical analyses. This study utilised stata survey commands (svy) to take into account the effects of clustering and unequal weights as appropriate when computing frequencies and confidence intervals. Rangpur division was formed in 2010 comprising of several districts of Rajshahi division. Hence, seven divisions were listed in BDHS 2011 and 2014. However, during the analyses, the authors merged this division with Rajshahi division and considered all the sampled observations in six divisions in all the four survey years. In BDHS surveys, the household wealth index was constructed from data on household assets, including ownership of durable goods (such as televisions and bicycles) and dwelling characteristics (such as source of drinking water, sanitation facilities, and construction materials), using principal component analysis [21].

To examine the central hypothesis of whether the place of residence modified the association between wealth and overweight/obesity, interaction terms between the place of residence and wealth index were fitted along with the main effects to measure the interaction effect for overweight and obesity in the logistic regression models for all the survey periods separately. However, during the analyses, the regression models were controlled with the relevant socio-demographic characteristics (educational status, watching television, current working status and use of contraceptive method) and all the results were reported with adjusted risk ratio (ARR) and 95% confidence interval. Before fitting the models, the potential collinearity of the predictor variables and the outcome variable were examined using the variance inflation factors.

As part of sensitivity analysis for the interaction effects of some other variables on overweight and obesity, this study investigated the interaction of both education and television viewing with the wealth index and with the place of residence separately. The analyses found no significant interaction effects on overweight or obesity in any of the surveys of the study sample for these variables. Hence, these interactions were not included in the final models.

**Ethical review.** The BDHS Data collection procedure was approved by the ICF Institutional Review Board (IRB) along with NIPORT, Mitra and Associates, the International Centre for Diarrhoeal Disease Research, Bangladesh, the Monitoring and Evaluation to Assess and Use the Results of Demographic and Health Surveys (MEASURE DHS) project funded by the United States Agency for International Development (USAID). All the DHS programs maintain strict standards to protect the privacy of respondents and household members, and the participation was voluntary, which was ensured by getting written consent from each participant.

## Results

### Socio-demographic characteristics and prevalence of obesity

The average age of women in the BDHS surveys across the years (2004, 20007, 2011 and 2014) was between 30.57 to 31.37 years (see Table 1). The cumulative prevalence of overweight and obesity, as shown in Table 1, was around 10% (overweight 7.44%, obese 2.52%) in 2004, while

**Table 1. Frequency distribution of BMI and socio-demographic characteristics of women of reproductive age—BDHS 2004, 2007, 2011 and 2014.**

| Background Characteristics | 2004 | | 2007 | | 2011 | | 2014 | |
|---|---|---|---|---|---|---|---|---|
| | n | % | n | % | n | % | n | % |
| **Body-Mass Index** | | | | | | | | |
| Non-Overweight/Obese | 9,385 | 90.04 | 8,727 | 87.02 | 13,387 | 81.59 | 12,546 | 75.57 |
| Overweight | 916 | 7.44 | 1,160 | 9.95 | 2,365 | 13.17 | 3,325 | 19.23 |
| Obese | 311 | 2.52 | 380 | 3.03 | 926 | 5.24 | 909 | 5.20 |
| **Age (n, mean)** | 10,612 | 30.57 | 10,267 | 30.96 | 16,678 | 31.23 | 16,780 | 31.37 |
| **Age Group** | | | | | | | | |
| 15–24 years | 3,340 | 31.80 | 3,039 | 30.30 | 4,670 | 28.74 | 4,476 | 27.05 |
| 25–34 years | 3,570 | 33.73 | 3,381 | 33.01 | 5,738 | 34.18 | 6,017 | 36.40 |
| 35–49 years | 3,702 | 34.47 | 3,847 | 36.69 | 6,270 | 37.08 | 6,287 | 36.56 |
| **Education Status** | | | | | | | | |
| No Education | 4,223 | 42.63 | 3,380 | 34.96 | 4,478 | 28.47 | 4,089 | 25.73 |
| Primary | 3,103 | 29.01 | 3,051 | 29.63 | 4,977 | 30.00 | 4,914 | 29.18 |
| Secondary | 2,647 | 23.41 | 3,029 | 29.48 | 5,858 | 34.24 | 6,192 | 36.69 |
| Higher | 639 | 4.96 | 807 | 5.93 | 1,365 | 7.29 | 1,585 | 8.41 |
| **Residence Status** | | | | | | | | |
| Rural | 6,979 | 77.40 | 6,364 | 77.29 | 10,821 | 73.71 | 10,952 | 71.56 |
| Urban | 3,633 | 22.60 | 3,903 | 22.71 | 5,857 | 26.29 | 5,828 | 28.44 |
| **Wealth by Residence Status** | | | | | | | | |
| *Rural* | | | | | | | | |
| Poorest | 1,592 | 23.32 | 1441 | 23.36 | 2,440 | 22.81 | 2,530 | 23.28 |
| Poorer | 1,546 | 23.00 | 1505 | 22.98 | 2,666 | 24.44 | 2,752 | 24.35 |
| Middle | 1,517 | 21.79 | 1465 | 22.44 | 2,578 | 23.95 | 2,626 | 23.47 |
| Richer | 1,510 | 21.24 | 1303 | 21.51 | 2,083 | 19.57 | 1,989 | 19.11 |
| Richest | 814 | 10.65 | 650 | 9.71 | 1,054 | 9.23 | 1,055 | 9.80 |
| *Urban* | | | | | | | | |
| Poorest | 310 | 8.60 | 217 | 4.99 | 431 | 5.25 | 492 | 6.57 |
| Poorer | 365 | 9.75 | 345 | 7.27 | 404 | 5.42 | 387 | 5.40 |
| Middle | 457 | 12.51 | 465 | 10.29 | 619 | 8.99 | 783 | 11.36 |
| Richer | 604 | 16.84 | 768 | 18.59 | 1473 | 24.35 | 1569 | 26.35 |
| Richest | 1899 | 52.30 | 2108 | 58.86 | 2930 | 56.00 | 2597 | 50.31 |
| **Division** | | | | | | | | |
| Barisal | 1,059 | 6.23 | 1,357 | 6.25 | 2,330 | 11.54 | 2,405 | 11.59 |
| Chittagong | 1,899 | 17.66 | 1,798 | 18.24 | 2,682 | 18.04 | 2,675 | 18.35 |
| Dhaka | 1,267 | 6.33 | 1,344 | 6.00 | 1,917 | 5.63 | 2,006 | 6.21 |
| Khulna | 1,593 | 12.25 | 1,612 | 12.80 | 2,525 | 12.27 | 2,470 | 10.46 |
| Rajshahi | 2,401 | 31.24 | 2,203 | 31.44 | 4,771 | 37.51 | 4,826 | 41.47 |
| Sylhet | 2,393 | 26.29 | 1,953 | 25.28 | 2,453 | 15.01 | 2,398 | 11.92 |
| **Marital Status** | | | | | | | | |
| Married | 9,732 | 91.97 | 9,419 | 92.21 | 15,546 | 93.32 | 15,747 | 94.01 |
| Widowed | 494 | 4.64 | 466 | 4.36 | 644 | 3.69 | 623 | 3.71 |
| Divorced/not living together | 386 | 3.39 | 382 | 3.43 | 488 | 2.99 | 410 | 2.27 |
| **Parity** | | | | | | | | |
| 0 | 1,035 | 9.44 | 955 | 8.91 | 1,426 | 8.67 | 1,412 | 8.04 |
| 1 | 2,070 | 19.02 | 2,183 | 21.29 | 3,627 | 21.53 | 3,910 | 23.33 |
| 2 | 2,570 | 24.15 | 2,640 | 25.72 | 4,813 | 28.60 | 5,020 | 30.10 |
| 3 | 2,002 | 19.52 | 1,958 | 19.85 | 3,351 | 20.28 | 3,306 | 20.07 |

*(Continued)*

**Table 1.** (Continued)

| Background Characteristics | 2004 | | 2007 | | 2011 | | 2014 | |
|---|---|---|---|---|---|---|---|---|
| | n | % | n | % | n | % | n | % |
| > = 4 | 2,935 | 27.87 | 2,531 | 24.24 | 3,461 | 20.92 | 3,132 | 18.45 |
| **Watching Television** | | | | | | | | |
| Not at all | 4,602 | 45.45 | 4,566 | 46.24 | 6,355 | 39.24 | 6,611 | 40.19 |
| Less than once a week | 947 | 9.25 | 648 | 7.01 | 1,991 | 12.28 | 1,465 | 8.64 |
| At least once a week | 5,063 | 45.30 | 5,053 | 46.75 | 8,332 | 48.48 | 8,704 | 51.17 |
| **Currently Working** | | | | | | | | |
| No | 8,190 | 76.82 | 7,174 | 67.18 | 14,386 | 86.47 | 11,338 | 65.89 |
| Yes | 2,422 | 23.18 | 3,093 | 32.82 | 2,292 | 13.53 | 5,442 | 34.11 |
| **Contraceptive Use** | | | | | | | | |
| Not using | 4,498 | 41.92 | 4,646 | 44.28 | 6,442 | 38.96 | 6,313 | 37.32 |
| Hormonal | 3,826 | 37.04 | 3,611 | 36.84 | 6,660 | 40.20 | 6,990 | 41.91 |
| Non-hormonal | 1,134 | 10.37 | 1,135 | 10.63 | 2,018 | 11.71 | 2,137 | 12.34 |
| Traditional | 1,154 | 10.67 | 875 | 8.25 | 1,558 | 9.13 | 1,340 | 8.44 |

it rose to 24% in 2014 (overweight 19.23%, obese 5.20%). The trend in the prevalence revealed a two-fold increase of both overweight and obesity in women over the study period, 2004–2014. Over 90% of the sample were married, and around three-quarters were living in rural areas. The distribution of wealth by place of residence remained almost similar over time with a greater concentration of richer and richest women among the residents of urban areas. Under the wealth index, in a rural setting, around 1 in every 4 women was from a very poor household, and 1 in every 10 women was from a very rich household; while in the urban settings, around 1 in every 2 women was from the richest households.

Tables 2 and 3 show the changes over time in the prevalence of overweight and obesity by socio-demographic characterics in women. The prevalence of both overweight and obesity was significantly greater in all subgroups in the 2014 sample. If we consider the increases of both overweight and obesity, as shown in Fig 1, the larger absolute increases in prevalence over time occurred in those who were in the age group of 25 to 34 and 35 to 49 years, married, having 2/3 children, watching television at least once a week and using non-hormonal contraceptive methods. There was a notable small increase in those who were in the age group of 15 to 24 years, had no children, never watched television and used hormonal contraceptive methods (see Fig 1).

Fig 2 depicts the changes over time in the prevalence of overweight and obesity in each wealth group of urban and rural women over the four survey periods. In both urban and rural households, overweight and obesity prevalence is higher in richest families compared to poorest households, and over time it increases significantly, especially in urban households. Fig 3 shows the increasing overweight and obesity prevalence trends among the women of reproductive age over the years by place of residence and division.

## Risk factors of overweight and obesity and interaction results

The adjusted associations between each of the socio-demographic characteristics and overweight or obesity, generated from the multinomial logistic regression models fitted to the overweight and obese category compared with BMI<25, are presented in Table 4. Overall, the risk factors of overweight and obesity differed among the survey years and also among the different sub-groups of socio-demographic characteristics. Increased age, higher education and higher

**Table 2. Prevalence of overweight and obesity by household socio-demographic characteristics in women (BDHS: 2004 and 2007).**

| Background Characteristics | 2004 | | | 2007 | | |
|---|---|---|---|---|---|---|
| | BMI <25 | Overweight | Obese | BMI <25 | Overweight | Obese |
| | % (95% CI) | % (95% CI) | % (95% CI) | % (95% CI) | % (95% CI) | % (95% CI) |
| **Age Group (years)** | | | | | | |
| 15–24 | 94.77 (93.94–95.50) | 3.63 (3.06–4.31) | 1.60 (1.21–2.11) | 93.25 (92.22–94.16) | 4.77 (4.01–5.67) | 1.97 (1.50–2.59) |
| 25–34 | 89.40 (88.20–90.49) | 7.92 (7.06–8.88) | 2.68 (2.12–3.38) | 84.88 (83.19–86.44) | 12.21 (10.92–13.64) | 2.9 (2.30–3.66) |
| 35–49 | 86.31 (84.47–87.95) | 10.48 (9.17–11.96) | 3.21 (2.57–4.00) | 83.78 (82.02–85.41) | 12.19 (10.80–13.73) | 4.02 (3.43–4.72) |
| **Education Status** | | | | | | |
| No Education | 93.76 (92.89–94.52) | 4.28 (3.67–4.98) | 1.97 (1.55–2.49) | 91.38 (90.24–92.40) | 6.06 (5.21–7.03) | 2.56 (2.00–3.28) |
| Primary | 91.15 (89.87–92.28) | 6.60 (5.64–7.71) | 2.25 (1.72–2.94) | 89.30 (87.8–90.63) | 8.48 (7.25–9.90) | 2.22 (1.73–2.84) |
| Secondary | 85.87 (84.16–87.42) | 10.91 (9.63–12.33) | 3.23 (2.49–4.17) | 83.52 (81.61–85.28) | 12.67 (11.2–14.29) | 3.81 (3.19–4.54) |
| Higher | 71.34 (65.61–76.45) | 23.19 (19.25–27.66) | 5.47 (3.48–8.52) | 67.25 (62.79–71.42) | 26.73 (23.37–30.39) | 6.01 (4.16–8.61) |
| **Marital Status** | | | | | | |
| Married | 90.02 (89.09–90.88) | 7.55 (6.86–8.30) | 2.43 (2.06–2.86) | 86.92 (85.82–87.95) | 10.14 (9.27–11.08) | 2.94 (2.55–3.39) |
| Widowed | 90.60 (87.36–93.07) | 7.33 (5.28–10.10) | 2.07 (1.16–3.66) | 87.16 (82.89–90.49) | 8.55 (6.07–11.93) | 4.29 (2.71–6.71) |
| Divorced/not living together | 89.75 (85.83–92.68) | 4.66 (2.93–7.31) | 5.59 (3.51–8.81) | 89.40 (85.92–92.10) | 6.68 (4.3–10.24) | 3.93 (2.10–7.21) |
| **Parity** | | | | | | |
| 0 | 91.41 (88.97–93.34) | 6.05 (4.48–8.13) | 2.54 (1.69–3.81) | 88.93 (86.52–90.95) | 6.99 (5.54–8.80) | 4.07 (2.86–5.76) |
| 1 | 91.38 (89.82–92.72) | 6.06 (5.01–7.30) | 2.56 (1.86–3.52) | 88.01 (86.12–89.68) | 9.12 (7.73–10.74) | 2.86 (2.13–3.84) |
| 2 | 88.92 (87.33–90.33) | 8.47 (7.36–9.72) | 2.61 (1.95–3.49) | 84.70 (82.97–86.29) | 12.40 (11.00–13.93) | 2.90 (2.26–3.73) |
| 3 | 88.69 (86.71–90.41) | 8.90 (7.44–10.62) | 2.41 (1.75–3.31) | 86.52 (84.59–88.24) | 10.88 (9.22–12.81) | 2.60 (2.00–3.37) |
| >= 4 | 90.58 (89.31–91.72) | 6.94 (5.99–8.04) | 2.47 (1.91–3.19) | 88.30 (86.31–90.03) | 8.41 (6.99–10.08) | 3.29 (2.55–4.24) |
| **Watching Television** | | | | | | |
| Not at all | 94.66 (93.89–95.33) | 3.22 (2.66–3.88) | 2.12 (1.69–2.68) | 92.05 (91.01–92.97) | 5.74 (4.98–6.60) | 2.21 (1.73–2.83) |
| Less than once a week | 93.41 (91.58–94.87) | 5.05 (3.74–6.80) | 1.53 (0.90–2.60) | 92.19 (89.34–94.32) | 7.13 (5.08–9.92) | 0.69 (0.27–1.75) |
| At least once a week | 84.72 (83.23–86.10) | 12.16 (11.09–13.33) | 3.11 (2.54–3.81) | 81.27 (79.78–82.67) | 14.54 (13.39–15.78) | 4.19 (3.57–4.92) |
| **Currently Working** | | | | | | |
| No | 89.40 (88.36–90.35) | 7.89 (7.14–8.71) | 2.71 (2.30–3.19) | 85.46 (84.13–86.69) | 11.19 (10.17–12.30) | 3.36 (2.87–3.91) |
| Yes | 92.18 (90.94–93.26) | 5.94 (4.93–7.15) | 1.88 (1.30–2.71) | 90.21 (88.81–91.45) | 7.42 (6.34–8.67) | 2.37 (1.84–3.05) |
| **Contraceptive Use** | | | | | | |
| Not using | 90.68 (89.42–91.80) | 6.50 (5.59–7.54) | 2.82 (2.32–3.43) | 87.71 (86.23–89.05) | 8.58 (7.44–9.88) | 3.71 (3.10–4.43) |
| Hormonal | 91.67 (90.63–92.60) | 6.70 (5.89–7.61) | 1.63 (1.21–2.19) | 88.79 (87.50–89.97) | 9.37 (8.32–10.55) | 1.83 (1.41–2.38) |
| Non-hormonal | 84.62 (81.63–87.20) | 11.57 (9.46–14.07) | 3.82 (2.72–5.33) | 80.91 (78.3–83.28) | 14.29 (12.27–16.58) | 4.79 (3.51–6.51) |
| Traditional | 87.17 (84.89–89.15) | 9.70 (8.00–11.73) | 3.13 (2.13–4.56) | 83.22 (79.9–86.08) | 14.28 (11.60–17.46) | 2.50 (1.62–3.83) |

wealth were associated with higher overweight and obesity risks among the respondents of all four periods.

In 2004, the women in the 35–49 age group were 4.64 times and 3.35 times more likely to be overweight and obese, respectively compared with counterparts aged 15–24 years. In 2014, women of this older age group (35–49) were 2.82 times and 3.71 times more likely to be overweight and obese, respectively, compared to the young age (15–24) women. Thus, there was a consistent trend of women being more likely to be overweight and obese in all the four survey periods over the decades for the age groups of 25–34 years and 35–49 years, with higher odds for the latter age group (see Table 4). Further, the risk of overweight and obesity was increased for women with higher education, ranging from the odds of 1.33–1.98 for secondary education and 1.35–1.79 for higher education compared with no education for all the four survey periods.

**Table 3. Prevalence of overweight and obesity by household socio-demographic characteristics in women (BDHS: 2011 and 2014).**

| Background Characteristics | 2011 | | | 2014 | | |
|---|---|---|---|---|---|---|
| | BMI <25 | Overweight | Obese | BMI <25 | Overweight | Obese |
| **Age Group (years)** | | | | | | |
| 15–24 | 89.80 (88.73–90.77) | 6.65 (5.89–7.51) | 3.55 (2.98–4.22) | 86.18 (84.68–87.55) | 11.13 (9.92–12.47) | 2.69 (2.17–3.33) |
| 25–34 | 78.93 (77.37–80.41) | 15.70 (14.38–17.12) | 5.37 (4.66–6.18) | 72.41 (70.35–74.37) | 22.39 (20.76–24.11) | 5.20 (4.49–6.03) |
| 35–49 | 77.68 (76.11–79.17) | 15.88 (14.74–17.09) | 6.44 (5.69–7.27) | 70.88 (68.81–72.87) | 22.08 (20.54–23.70) | 7.04 (6.11–8.10) |
| **Education Status** | | | | | | |
| No Education | 87.16 (85.90–88.32) | 9.07 (8.11–10.13) | 3.78 (3.16–4.51) | 83.37 (81–85.49) | 13.05 (11.24–15.10) | 3.58 (2.92–4.40) |
| Primary | 84.49 (83.08–85.79) | 11.17 (10.04–12.42) | 4.34 (3.73–5.05) | 78.69 (77.11–80.18) | 17.00 (15.74–18.33) | 4.32 (3.66–5.09) |
| Secondary | 78.41 (76.85–79.90) | 15.59 (14.45–16.81) | 5.99 (5.24–6.85) | 71.40 (69.48–73.24) | 22.55 (21.11–24.06) | 6.05 (5.13–7.14) |
| Higher | 62.82 (59.21–66.30) | 26.01 (23.52–28.68) | 11.16 (8.65–14.28) | 59.16 (55.71–62.53) | 31.42 (28.40–34.60) | 9.42 (7.96–11.11) |
| **Marital Status** | | | | | | |
| Married | 81.54 (80.48–82.56) | 13.26 (12.44–14.12) | 5.20 (4.72–5.74) | 75.21 (73.64–76.73) | 19.60 (18.45–20.80) | 5.19 (4.61–5.84) |
| Widowed | 81.68 (77.66–85.12) | 12.50 (9.70–15.96) | 5.82 (3.98–8.42) | 79.52 (74.82–83.54) | 15.23 (12.00–19.14) | 5.24 (3.43–7.95) |
| Divorced/not living together | 82.95 (77.89–87.04) | 11.29 (8.20–15.35) | 5.76 (3.61–9.06) | 83.96 (79.45–87.64) | 10.61 (7.52–14.78) | 5.43 (3.31–8.77) |
| **Parity** | | | | | | |
| 0 | 84.14 (81.73–86.28) | 10.13 (8.49–12.05) | 5.73 (4.45–7.36) | 81.37 (78.73–83.75) | 13.34 (11.32–15.65) | 5.29 (3.92–7.11) |
| 1 | 83.63 (81.99–85.14) | 11.48 (10.28–12.81) | 4.89 (4.01–5.94) | 79.32 (77.04–81.43) | 16.59 (14.79–18.56) | 4.09 (3.43–4.86) |
| 2 | 78.37 (76.84–79.82) | 15.98 (14.66–17.38) | 5.66 (4.87–6.56) | 70.91 (68.82–72.91) | 22.98 (21.37–24.67) | 6.11 (5.22–7.15) |
| 3 | 79.92 (78.18–81.54) | 14.73 (13.28–16.31) | 5.35 (4.57–6.26) | 73.69 (70.79–76.39) | 20.59 (18.39–22.98) | 5.72 (4.74–6.88) |
| > = 4 | 84.46 (82.8–85.98) | 10.81 (9.58–12.18) | 4.74 (4.02–5.57) | 77.97 (76.00–79.83) | 17.54 (15.93–19.28) | 4.49 (3.67–5.47) |
| **Watching Television** | | | | | | |
| Not at all | 89.96 (88.94–90.89) | 7.09 (6.32–7.95) | 2.95 (2.48–3.50) | 85.72 (84.24–87.08) | 11.72 (10.54–13.00) | 2.56 (2.05–3.20) |
| Less than once a week | 85.76 (83.86–87.46) | 10.09 (8.67–11.71) | 4.15 (3.28–5.24) | 80.54 (78.18–82.70) | 16.01 (13.96–18.29) | 3.45 (2.54–4.67) |
| At least once a week | 73.76 (72.18–75.27) | 18.87 (17.70–20.10) | 7.38 (6.53–8.32) | 66.77 (64.94–68.54) | 25.68 (24.33–27.07) | 7.56 (6.71–8.50) |
| **Currently Working** | | | | | | |
| No | 81.88 (80.74–82.95) | 13.01 (12.16–13.90) | 5.12 (4.60–5.69) | 73.58 (71.77–75.31) | 20.33 (19.07–21.65) | 6.09 (5.35–6.94) |
| Yes | 79.75 (77.64–81.72) | 14.20 (12.57–16.00) | 6.05 (4.92–7.41) | 79.43 (77.64–81.11) | 17.11 (15.62–18.71) | 3.46 (2.86–4.18) |
| **Contraceptive Use** | | | | | | |
| Not using | 81.35 (79.87–82.75) | 12.86 (11.77–14.04) | 5.79 (5.08–6.58) | 75.62 (73.62–77.52) | 18.54 (16.99–20.20) | 5.83 (5.11–6.65) |
| Hormonal | 84.01 (82.77–85.17) | 12.13 (11.10–13.25) | 3.86 (3.36–4.42) | 78.98 (77.54–80.35) | 17.47 (16.33–18.68) | 3.55 (2.97–4.25) |
| Non-hormonal | 75.20 (72.60–77.63) | 17.52 (15.68–19.54) | 7.27 (5.97–8.84) | 68.11 (64.82–71.23) | 24.72 (22.18–27.44) | 7.17 (5.82–8.80) |
| Traditional | 80.15 (77.43–82.62) | 13.44 (11.43–15.75) | 6.4 (5.09–8.03) | 69.36 (64.45–73.87) | 22.99 (19.56–26.81) | 7.65 (5.83–9.98) |

In terms of wealth index, the richest women had 4.11 and 4.13 higher odds of being overweight compared with women in the poorest wealth index group in 2004 and 2014 respectively. Though there were fluctuations in the magnitude of odds of being obese for both richer and richest women, there were consistent trends in the odds of being overweight for the women of these wealth groups in all four time periods. The poorer women were not significantly associated with the risk of being overweight or obese.

Marital status was not significantly associated with overweight or obesity for the years 2004, 2007 and 2011. In 2014, widowed or divorced women were considerably less likely to be overweight or obese compared with married women. Further, residence status was not significantly associated with overweight or obesity across all survey years. Watching television at least once a week was a risk factor in selected BDHS surveys for overweight (with the odds of 2.01,1.43, and 1.32 for 2004, 2011 and 2014 respectively) and obesity (with the odds of 1.46 and 1.43 for 2011 and 20014 respectively), compared with the women who did not watch television at all.

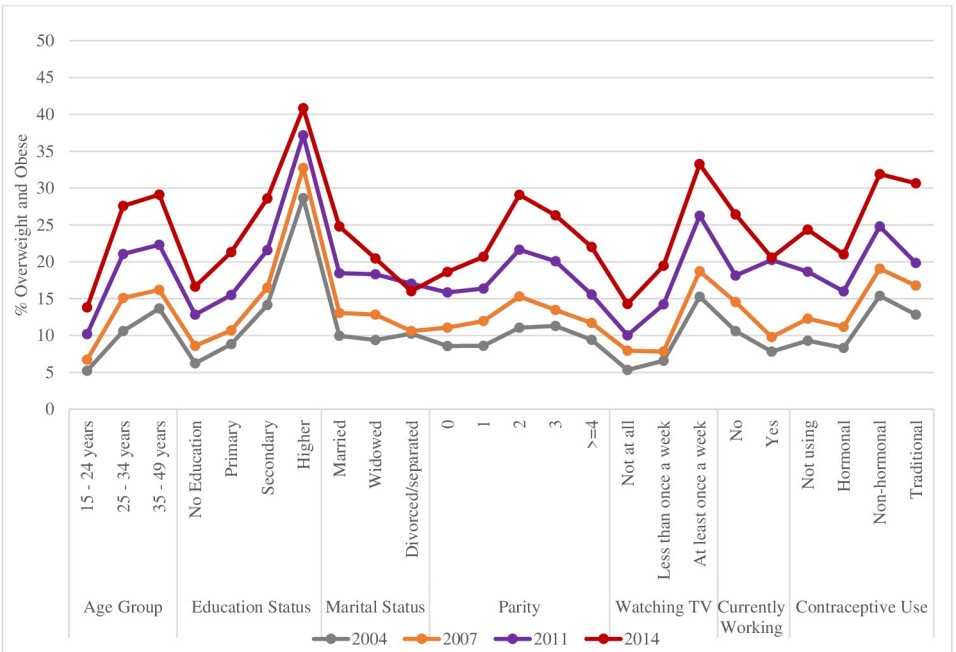

**Fig 1. Prevalence of sum of overweight and obese women aged 15–49 years over the survey periods by socio-demographic characteristics–BDHS 2004, 2007, 2011, 2014.**

Working women were less likely to be overweight and obese in all three survey time points, except for the 2011 survey year which did not show any association in this regard.

The interaction between the place of residence and wealth index showed that the place of residence modified the association between wealth and obesity significantly, but not

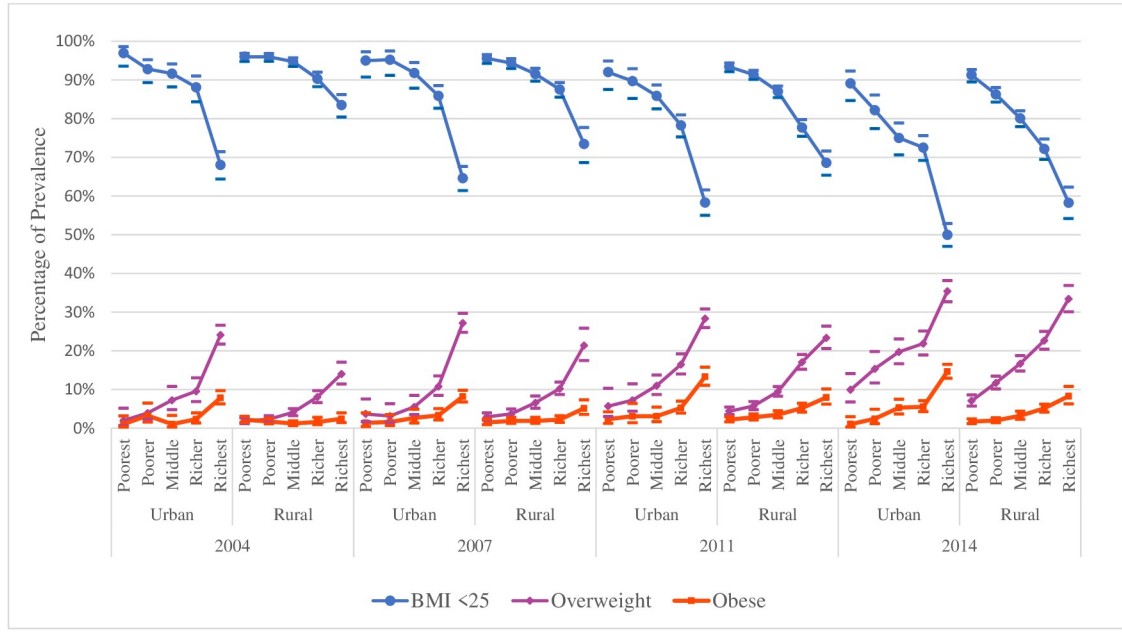

**Fig 2. BMI status of women aged 15–49 years over the survey periods by type of residence and wealth index—BDHS 2004, 2007, 2011 and 2014.**

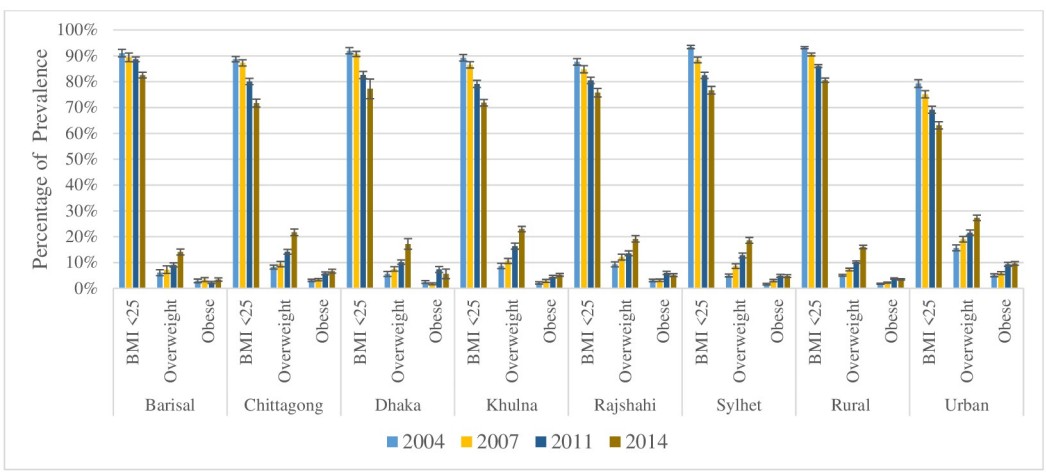

**Fig 3. BMI status of women aged 15–49 years over the survey periods by division or place of residence—BDHS 2004, 2007, 2011, 2014.**

overweight. The study result revealed that the urban richer and richest women were respectively 4.23 and 5.99 times more likely to be obese in 2004. In comparison, the study observed a reduced magnitude of odds ratio in 2014 where urban middle income and richest women were respectively 3.00 times and 2.94 times more likely to be obese, compared to their counterparts in rural areas.

## Discussion

The present study articulates that the prevalence of overweight and obesity among women aged 15–49 years has considerably increased from 9.96% in 2004 to 24.43% in 2014. Using multinomial logistic regression, this study found a significant multiplicative interaction between wealth and place of residence with an increased risk of obesity in 2004 and 2014 BDHS surveys. Moreover, the present study found that higher wealth, higher education, older age and increased television viewing were positively associated and the working status of women was negatively associated with overweight and obesity in women in all survey periods.

This study further revealed that the place of residence appeared to be a risk factor of obesity among women belonging to the richest wealth quantile. Women living in urban settings, moving up to the richest wealth quantile were associated with higher odds of being obese in the 2004 and 2014 BDHS survey periods. However, the place of residence alone was not associated with the increased risk of overweight or obesity among women.

Previous studies separately identified wealth and place of living as risk factors of obesity among women. There is limited literature to compare the interaction results as no study investigated the association between wealth and place of living with the risk of obesity. However, prior studies on Bangladeshi women confirmed that the odds of being obese are higher among women from the richest households living in urban settings (3, 6). One possible explanation might be that wealthier people in urban settings lead a sedentary lifestyle, do less physical activities, and engage in less labour-intensive occupations along with the consumption of energy-dense foods (18), compared with their rural counterparts. Another possible explanation is that wealth gives more access to food and escape from physical labour (17), and rapid urbanisation contributes to overweight and obesity by providing more access to technologies that require less energy, together with the availability of high-calorie foods, and limited space for physical activities.

**Table 4. Multinomial logistic regression models for the association of overweight and obesity with socio-demographic characteristics and the interaction of the place of residence and wealth of women at four-time points (BDHS Surveys: 2004, 2007, 2011 and 2014).**

| Sociodemographic factors | 2004 | | 2007 | | 2011 | | 2014 | |
|---|---|---|---|---|---|---|---|---|
| | Overweight | Obese | Overweight | Obese | Overweight | Obese | Overweight | Obese |
| | RRR (95% CI), *p*-value | RRR (95% CI), *p*-value | RRR (95% CI), *p-value* | RRR (95% CI), *p*-value | RRR (95% CI), *p*-value | RRR (95% CI), *p*-value | RRR (95% CI), *p*-value | RRR (95% CI), *p*-value |
| **Age group** | | | | | | | | |
| 15–24 (Ref.) | | | | | | | | |
| 25–34 | **2.83 (2.22–3.60), <0.001** | **2.24 (1.55–3.25), <0.001** | **3.44 (2.76–4.30), <0.001** | **2.55 (1.80–3.62), <0.001** | **2.59 (2.22–3.02), <0.001** | **1.87 (1.50–2.34), <0.001** | **2.39 (2.09–2.72), <0.001** | **2.57 (2.02–3.27), <0.001** |
| 35–49 | **4.64 (3.55–6.06), <0.001** | **3.35 (2.23–5.02), <0.001** | **4.42 (3.45–5.65), <0.001** | **3.53 (2.41–5.17), <0.001** | **3.29 (2.77–3.91), <0.001** | **2.63 (2.06–3.37), <0.001** | **2.82 (2.42–3.27), <0.001** | **3.71 (2.85–4.84), <0.001** |
| **Education Level** | | | | | | | | |
| No education (Ref.) | | | | | | | | |
| Primary | **1.36 (1.10–1.68), 0.005** | **1.54 (1.11–2.15), 0.011** | **1.40 (1.14–1.71), 0.001** | 1.10 (0.80–1.53), 0.549 | 1.14 (0.99–1.32), 0.065 | 1.13 (0.91–1.40), 0.262 | **1.25 (1.10–1.41), <0.001** | 1.21 (0.97–1.51), 0.092 |
| Secondary | **1.98 (1.59–2.46), <0.001** | **2.41 (1.68–3.45), <0.001** | **1.89 (1.53–2.32), <0.001** | **1.87 (1.35–2.59), <0.001** | **1.36 (1.17–1.57), <0.001** | **1.41 (1.13–1.76), 0.003** | **1.50 (1.32–1.71), <0.001** | **1.33 (1.06–1.67), 0.014** |
| Higher | **2.71 (2.03–3.61), <0.001** | **2.79 (1.73–4.52), <0.001** | **2.50 (1.93–3.25), <0.001** | **1.74 (1.14–2.65), 0.010** | **1.52 (1.25–1.85), <0.001** | **1.78 (1.35–2.36), <0.001** | **1.63 (1.37–1.94), <0.001** | **1.35 (1.01–1.80), 0.042** |
| **Residence** | | | | | | | | |
| Rural (Ref.) | | | | | | | | |
| urban | 0.99 (0.40–2.42), 0.979 | 0.56 (0.20–1.60), 0.281 | 1.35 (0.62–2.94), 0.448 | 0.92 (0.27–3.12), 0.897 | 0.90 (0.55–1.49), 0.696 | 1.16 (0.61–2.19), 0.651 | **1.39 (1.00–1.91), 0.047** | 0.70 (0.30–1.66), 0.417 |
| **Wealth Index** | | | | | | | | |
| Poorest (Ref.) | | | | | | | | |
| poorer | 0.97 (0.57–1.64), 0.911 | 0.68 (0.41–1.13), 0.133 | 1.29 (0.85–1.95), 0.227 | 1.15 (0.66–2.02), 0.621 | 1.18 (0.92–1.53), 0.193 | 1.28 (0.90–1.80), 0.165 | **1.33 (1.10–1.61), 0.004** | 1.21 (0.81–1.79), 0.356 |
| middle | **1.67 (1.04–2.67), 0.032** | **0.41 (0.23–0.73), 0.003** | **2.00 (1.36–2.94), <0.001** | 0.94 (0.53–1.68), 0.836 | **1.82 (1.43–2.31), <0.001** | 1.25 (0.88–1.77), 0.214 | **1.72 (1.42–2.08), <0.001** | 1.38 (0.93–2.06), 0.109 |
| richer | **2.61 (1.68–4.06), <0.001** | **0.45 (0.26–0.79), 0.005** | **2.70 (1.84–3.96), <0.001** | 1.12 (0.63–1.99), 0.708 | **3.02 (2.37–3.84), <0.001** | **1.87 (1.31–2.67), 0.001** | **2.64 (2.16–3.23), <0.001** | **2.72 (1.83–4.03), <0.001** |
| richest | **4.11 (2.59–6.52), <0.001** | 0.83 (0.46–1.48), 0.523 | **5.60 (3.75–8.36), <0.001** | **2.29 (1.27–4.15), 0.006** | **4.29 (3.28–5.60), <0.001** | **2.93 (1.99–4.32), <0.001** | **4.13 (3.30–5.18), <0.001** | **4.44 (2.90–6.80), <0.001** |
| **Division** | | | | | | | | |
| Dhaka (Ref.) | | | | | | | | |
| Chittagong | 1.25 (0.94–1.66), 0.131 | 1.07 (0.70–1.63), 0.770 | 1.06 (0.82–1.37), 0.661 | 1.37 (0.89–2.11), 0.150 | 1.05 (0.87–1.26), 0.635 | **0.70 (0.55–0.90), 0.005** | 1.15 (0.98–1.35), 0.084 | 1.24 (0.94–1.64), 0.131 |
| Rajshahi | 1.32 (1.01–1.74), 0.046 | 1.06 (0.70–1.59), 0.790 | 1.10 (0.86–1.41), 0.432 | 1.06 (0.69–1.62), 0.805 | 0.83 (0.70–0.99), 0.036 | **0.58 (0.46–0.72), <0.001** | 0.88 (0.76–1.02), 0.095 | 0.87 (0.67–1.13), 0.302 |
| Khulna | 1.23 (0.92–1.65), 0.159 | 0.92 (0.59–1.45), 0.722 | 1.14 (0.88–1.47), 0.324 | 1.24 (0.79–1.94), 0.351 | 1.23 (1.02–1.48), 0.026 | **0.59 (0.46–0.76), <0.001** | 1.27 (1.08–1.49), 0.004 | 1.25 (0.94–1.66), 0.132 |
| Sylhet | 1.02 (0.76–1.36), 0.919 | 0.65 (0.41–1.03), 0.069 | 1.15 (0.89–1.48), 0.290 | 1.51 (0.99–2.31), 0.056 | 1.07 (0.89–1.30), 0.466 | **0.64 (0.49–0.82), 0.001** | 1.13 (0.96–1.33), 0.154 | 1.23 (0.92–1.64), 0.163 |
| Barisal | 1.09 (0.78–1.54), 0.617 | 1.16 (0.72–1.88), 0.537 | 0.79 (0.59–1.06), 0.118 | 1.28 (0.81–2.02), 0.284 | 0.90 (0.74–1.10), 0.310 | **0.38 (0.28–0.52), <0.001** | 0.94 (0.79–1.12), 0.495 | 0.99 (0.73–1.35), 0.949 |
| **Marital Status** | | | | | | | | |
| Married (Ref.) | | | | | | | | |
| Widowed | 1.01 (0.70–1.46), 0.945 | 0.85 (0.48–1.50), 0.580 | 0.88 (0.62–1.26), 0.491 | 1.42 (0.92–2.19), 0.115 | 0.91 (0.71–1.17), 0.447 | 1.00 (0.70–1.42), 0.998 | **0.72 (0.57–0.91), 0.006** | 0.78 (0.54–1.12), 0.181 |
| Divorced/not living together | 0.78 (0.48–1.29), 0.339 | 1.66 (0.95–2.88), 0.074 | 0.73 (0.47–1.12), 0.153 | 0.94 (0.53–1.65), 0.822 | 0.73 (0.52–1.01), 0.058 | 1.05 (0.69–1.59), 0.820 | **0.51 (0.36–0.72), <0.001** | 0.96 (0.61–1.52), 0.869 |
| **Parity** | | | | | | | | |
| 0 (Ref.) | | | | | | | | |

*(Continued)*

**Table 4.** (Continued)

| Sociodemographic factors | 2004 | | 2007 | | 2011 | | 2014 | |
|---|---|---|---|---|---|---|---|---|
| | Overweight | Obese | Overweight | Obese | Overweight | Obese | Overweight | Obese |
| | RRR (95% CI), *p*-value | RRR (95% CI), *p*-value | RRR (95% CI), *p-value* | RRR (95% CI), *p*-value | RRR (95% CI), *p*-value | RRR (95% CI), *p*-value | RRR (95% CI), *p*-value | RRR (95% CI), *p*-value |
| 1 | 1.04 (0.75–1.43), 0.831 | 1.33 (0.82–2.17), 0.246 | 0.92 (0.69–1.23), 0.583 | 0.68 (0.45–1.01), 0.055 | 1.03 (0.83–1.27), 0.798 | 0.89 (0.67–1.18), 0.413 | **1.23 (1.02–1.48), 0.030** | 1.00 (0.74–1.36), 0.996 |
| 2 | 1.07 (0.77–1.48), 0.677 | 1.24 (0.75–2.04), 0.400 | 0.99 (0.74–1.33), 0.947 | **0.64 (0.42–0.97), 0.037** | 1.11 (0.89–1.37), 0.357 | 0.91 (0.68–1.22), 0.528 | **1.43 (1.17–1.73), <0.001** | 1.11 (0.81–1.53), 0.507 |
| 3 | 1.07 (0.76–1.53), 0.692 | 1.18 (0.68–2.03), 0.557 | 0.85 (0.62–1.17), 0.320 | **0.54 (0.34–0.85), 0.008** | 1.01 (0.80–1.27), 0.963 | 0.83 (0.60–1.15), 0.258 | **1.32 (1.07–1.63), 0.010** | 1.08 (0.76–1.52), 0.673 |
| > = 4 | 0.93 (0.65–1.33), 0.689 | 1.10 (0.64–1.91), 0.726 | 0.75 (0.54–1.04), 0.082 | **0.60 (0.38–0.95), 0.031** | 0.89 (0.70–1.14), 0.353 | 0.85 (0.61–1.19), 0.343 | **1.21 (0.97–1.51), 0.095** | 0.88 (0.61–1.27), 0.493 |
| **Watching Television** | | | | | | | | |
| Not at all (Ref.) | | | | | | | | |
| Less than once a week | **1.42 (1.02–1.97), 0.040** | 0.70 (0.41–1.20), 0.194 | 0.85 (0.60–1.20), 0.356 | **0.47 (0.23–0.98), 0.044** | **1.20 (1.00–1.43), 0.050** | 1.25 (0.95–1.63), 0.108 | 1.13 (0.96–1.34), 0.132 | 1.05 (0.76–1.45), 0.760 |
| At least once a week | **2.01 (1.62–2.49), <0.001** | 0.90 (0.66–1.24), 0.528 | 1.15 (0.96–1.37), 0.130 | 1.30 (0.97–1.74), 0.080 | **1.43 (1.25–1.64), <0.001** | **1.46 (1.18–1.80), <0.001** | **1.32 (1.18–1.48), <0.001** | **1.43 (1.15–1.78), 0.001** |
| **Currently Working** | | | | | | | | |
| No (Ref.) | | | | | | | | |
| Yes | **0.80 (0.66–0.97), 0.023** | **0.64 (0.46–0.88), 0.006** | **0.67 (0.57–0.79), <0.001** | **0.67 (0.51–0.87), 0.003** | 0.99 (0.86–1.14), 0.909 | 0.99 (0.81–1.21), 0.937 | **0.89 (0.82–0.98), 0.013** | **0.63 (0.53–0.74), <0.001** |
| **Contraceptive use** | | | | | | | | |
| Not using (Ref.) | | | | | | | | |
| Hormonal | 0.92 (0.76–1.11), 0.377 | **0.53 (0.39–0.73), <0.001** | 1.00 (0.85–1.19), 0.984 | **0.59 (0.44–0.79), <0.001** | 0.93 (0.83–1.05), 0.243 | **0.76 (0.63–0.90), 0.002** | 0.91 (0.82–1.01), 0.068 | **0.64 (0.53–0.77), <0.001** |
| Non-hormonal | 0.94 (0.74–1.19), 0.588 | 0.97 (0.68–1.37), 0.853 | 1.06 (0.86–1.31), 0.596 | 1.14 (0.83–1.55), 0.421 | 1.15 (1.00–1.34), 0.057 | 1.12 (0.91–1.39), 0.294 | 1.01 (0.89–1.15), 0.872 | 0.93 (0.75–1.15), 0.499 |
| Traditional | 1.21 (0.96–1.54), 0.109 | 0.88 (0.60–1.29), 0.523 | 1.12 (0.88–1.41), 0.359 | 0.76 (0.50–1.14), 0.179 | 0.88 (0.74–1.05), 0.156 | 1.02 (0.80–1.29), 0.874 | 1.12 (0.96–1.31), 0.139 | 1.17 (0.92–1.49), 0.199 |
| **Interaction of Place of Residence and Wealth Index** | | | | | | | | |
| urban#poorer | 2.06 (0.69–6.12), 0.194 | 3.15 (0.88–11.33), 0.078 | 0.71 (0.26–1.94), 0.503 | 1.01 (0.22–4.58), 0.989 | 1.40 (0.73–2.66), 0.310 | 0.57 (0.22–1.47), 0.246 | 1.02 (0.66–1.59), 0.921 | 2.02 (0.69–5.88), 0.199 |
| urban#middle | 1.58 (0.58–4.33), 0.371 | 1.55 (0.34–7.04), 0.567 | 0.71 (0.29–1.73), 0.448 | 1.70 (0.42–6.92), 0.460 | 1.42 (0.80–2.51), 0.234 | 0.88 (0.39–1.96), 0.753 | 1.08 (0.73–1.58), 0.709 | **3.00 (1.16–7.74), 0.023** |
| urban#richer | 1.07 (0.41–2.81), 0.883 | **4.23 (1.25–14.34), 0.021** | 0.88 (0.38–2.03), 0.774 | 1.64 (0.43–6.22), 0.465 | 1.22 (0.71–2.07), 0.471 | 0.89 (0.44–1.79), 0.735 | 0.77 (0.54–1.11), 0.157 | 1.65 (0.66–4.09), 0.283 |
| urban#richest | 1.82 (0.72–4.60), 0.206 | **5.99 (1.91–18.74), 0.002** | 1.09 (0.48–2.44), 0.841 | 2.27 (0.63–8.15), 0.209 | 1.69 (1.00–2.88), 0.052 | 1.44 (0.73–2.87), 0.293 | 0.97 (0.68–1.39), 0.866 | **2.94 (1.20–7.24), 0.019** |

This study revealed that the obesity level increased in women who were in the richest quantile compared with women from the poorest wealth quantile in all the survey periods. The findings of the study concerning the wealth-obesity relationship are consistent with prior studies (6, 12). One possible explanation for the richest women having higher rates of obesity than the poorest women is changing in dietary behaviour with the changes in income. There is evidence that consumption of higher energy and fat, and processed food increases with higher income (17).

An earlier study showed that a higher level of education is a protective factor of obesity among women (16). The association between wealth and obesity is not inevitable and can be changed through investment in education (16). Taking it into consideration, this study suggests that initiatives should be undertaken to develop socio-culturally appropriate

guidelines to maintain a healthy weight and implement interventions for those who are already obese.

The present study has several notable strengths that should be mentioned. The main strength of the present study is the large sample size and its representativeness of the general population. This study utilised four large and nationally representative surveys that extensively cover both rural and urban settings in Bangladesh. One strength of the present study is that unlike previous studies in Bangladesh, this is the first study that has attempted to identify the association between wealth, place of living and their interaction with obesity. Another strength of this study is that BMI is the focus of the present study, and data on BMI were collected by trained interviewers following the internationally recommended standard protocols. The use of trained interviewers helps to capture the BMI of the study participants accurately, and thus this study has avoided self-reported bias.

## Conclusion

Overweight and obesity among women at reproductive age is a growing public health concern in Bangladesh. This study offered insights that urban wealthier women of reproductive age are more prone to be obese. However, the underlying reasons for the greater risk of urban wealthier women to be obese require further investigation. The present study has some limitations: first, the present study is unable to draw directional causal inferences as it is based on a cross-sectional design. Additionally, this study cannot incorporate some critical risk factors of obesity, such as respondents' dietary habits, physical activity, sedentary behaviour, and comorbid conditions, due to their unavailability in BDHS data sets. Future studies may recheck the association following prospective longitudinal research design where individuals will have been tracked and thus can capture the within-person change in BMI. Finally, we suggest that policymakers should undertake intervention programs targeting urban wealthier women who are more prone to be obese.

## Acknowledgments

The authors would like to thank the Demographic and Health Surveys (DHS) Program, funded by the United States Agency for International Development, for providing the Bangladesh DHS datasets. The authors would also like to thank Dr Barbara Harmes for proofreading the manuscript before submission.

## Author Contributions

**Conceptualization:** Kabir Ahmad.

**Data curation:** Kabir Ahmad.

**Formal analysis:** Kabir Ahmad, Taslima Khanam.

**Methodology:** Kabir Ahmad, Taslima Khanam.

**Software:** Kabir Ahmad.

**Supervision:** Enamul Kabir, Rasheda Khanam.

**Writing – original draft:** Kabir Ahmad, Taslima Khanam, Syed Afroz Keramat, Md. Irteja Islam.

**Writing – review & editing:** Kabir Ahmad, Enamul Kabir, Rasheda Khanam.

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
