## [Decision Letter · Decision Letter 0]

15 Sep 2020

PONE-D-20-25454

Interaction between Wealth and Place of Residence on the Risk of Obesity in Bangladeshi Women

PLOS ONE

Dear Authors,

Thank you for submitting your manuscript to PLOS ONE. After careful consideration, we feel that it has merit but does not fully meet PLOS ONE’s publication criteria as it currently stands. Therefore, we invite you to submit a revised version of the manuscript that addresses the points raised during the review process.

We look forward to receiving your revised manuscript.

Kind regards,

Professor Hafiz T.A. Khan, Ph.D, CStat

Academic Editor

PLOS ONE

Journal Requirements:

'This research did not receive any specific grant from funding agencies in the public, commercial or not-for-profit sectors'

We note that one or more of the authors are employed by a commercial company: Purple Informatics

4. Please ensure that you include a title page within your main document. We do appreciate that you have a title page document uploaded as a separate file, however, as per our author guidelines (http://journals.plos.org/plosone/s/submission-guidelines#loc-title-page) we do require this to be part of the manuscript file itself and not uploaded separately.

Reviewers' comments:

Reviewer's Responses to Questions

**Comments to the Author**

1. Is the manuscript technically sound, and do the data support the conclusions?

Reviewer #1: Yes

Reviewer #2: Partly

2. Has the statistical analysis been performed appropriately and rigorously? 

Reviewer #1: Yes

Reviewer #2: Yes

3. Have the authors made all data underlying the findings in their manuscript fully available?

Reviewer #1: Yes

Reviewer #2: Yes

4. Is the manuscript presented in an intelligible fashion and written in standard English?

Reviewer #1: Yes

Reviewer #2: Yes

5. Review Comments to the Author

Reviewer #1: This paper analyzed some specific determinants on prevalence of obesity among Bangladeshi women. Although the topic is interesting and useful in national level, the scope of the paper is reduced by a brief analysis. The analysis could bring important insights from the available data, but the Authors skipped that by summarizing the long time series data. In it's present form the paper is monotonous to attract readers and it may be benefited from some graphical representations of the results. Some of my queries and specific recommendations are given below.

1. There should be some statements in the rationale of the study regarding the negative long run impact of obesity, both from national and international point of view. Some references citing the statistical relation regarding complexities of obesity among women will do.

2. Query: already several studies are published regarding obesity in Bangladesh and the Authors also cited some of them. What extra information are being added by this paper or how the current paper is superior over those? The rationale for focusing on these particular variables/interactions of interest are missing in the Introduction to clarify that.

3. Choice of the independent variables requires brief but clearer explanation regarding the physiological/behavioral factors associated with obesity. The Authors should cover-up the existing vast literature more carefully to explain the scenario of obesity among Bangladeshi women.

4. Query: until BDHS 2014, huge proportion of the respondents used to listen radio on a regular basis. Specially during 2007~11, the new FM radio stations were popular in Bangladesh. Same concept applies for newspapers but for a more educated part of the population. Why did the Authors chose to consider only TV over mass media (by combining the joint effect of TV, Radio and newspaper together)? The Authors need to clarify this in the paper.

5. Pooling the samples from different BDHSs reduces the scope of this paper. Instead of showing the results for two distinct decades, results from each individual survey will show the trend and it will be more useful for policy making. The Authors should present the results separately for all four surveys.

6. Some other possible interactions in addition to the mentioned ones may have significant impact on obesity. I understand that consideration of the other interactions are outside of the scope for the current paper, however the Authors should mention some of those in the paper.

7. The result analysis should be stronger (see the next comments). This analysis contains important time series data for obesity trend among Bangladeshi women, but the Authors did not focus on that in the findings. This applies for all three types of findings: distribution, bivariate associations and regression analysis.

8. All the tables title need to be edited to mention the content properly. Data sources are also missing in the titles.

9. What does this CI stands for in Table 1? The Authors showed the frequency in that table, not average. I suggest to remove it.

10. Table 2 is difficult to understand for weak presentation of too many information (including title). The Authors need to change the captions and stubs properly. The absolute differences should be removed. However, I highly recommend to replace this table with specific graphical representation(s). In addition, the Authors need to include appropriate graphical representation(s) to sketch the relation between the interactions and prevalence of obesity.

11. In Table 3 and in it's description, the reference group should be mentioned clearly. Instead of present numeric results again in the adjacent texts, the Authors should discuss the key findings only without repeating the same numbers again.

12. The first sentence of the Discussion should be placed as the first sentence of the Conclusion. The other statements of the first paragraph of Discussion should be removed.

13. The Discussion should focus the key findings only and it may include some useful findings. Statistical terms should be explained more carefully to avoid ambiguity (specially in page 16). Any statement addressing the policy-makers should be removed.

14. The second paragraph of page 17 should be rewritten very briefly (not more than 2/3 sentences) according to modified analysis. The modified statements should be placed in the Introduction as usefulness of the analysis. Else, the Authors should remove this paragraph.

15. The last paragraph of the Discussion should be edited and placed in the Conclusion (check next comment).

16. Conclusion should contain a brief summary in 2/3 sentences; possible consequences in few words; limitation of the work followed by scopes of future research to overcome that; and how to extend the work. The Authors may suggest something to the policy-makers after fulfilling the structure but that's not a mandatory element.

17. The manuscript requires a careful edit for small typos and consistencies in citation style.

Reviewer #2: General comments

The manuscript can be legible for publication after addressing the comments raised. There are glaring grammatical errors. The authors need to address the comments and submit the manuscript for consideration in publication.

Comments to the author

Abstract

The title of the study indicates the focus on obesity only but from the data analysis it is clear that the authors are referring to overweight/obesity. I would suggest the title includes overweight because the authors are looking at this outcome also.

Line 28-In the conclusion section of the abstract use small ‘e’ for Education

Line 28- replace the word ‘is’ with ‘were’

There is a subsection on key messages in the abstract; I am wondering if this is consistent with PLOS One article writing format.

Introduction

The introduction provides background on overweight/obesity, starting at the global context and narrowing down to the Bangladeshi context. One thing I see lacking is what other studies have found on the interaction of wealth and place of residence on the risk of overweight/obesity. Are there any studies which have explored the interaction of these two socioeconomic variables? Where were these studies done? High income countries? LMICs?

Lines 59-66- The authors need to revise this section and put it in a clearer language. It is difficult to understand what the authors are saying.

Line 67, use small letter ‘o’ for the word ‘Overweight’

Line 69, remove the word ‘therefore’

Methodology

The methodology section is comprehensively written.

Line-116 Consider putting overweight/obesity in the sentence

Line 124- It is not clear from the statistical analyses section how the interaction between place of residence and household wealth was measured. Was it measured using the interaction effect???. Let it be clear how you analysed the interaction since this is the core of the paper.

Results

Line 144-145-This sentence need to be revised. It is not clear what the authors are saying

Table1- In Table 1 the totals (N) for background characteristics are not equal throughout. I expect this to be consistently the same considering the exclusion criteria which indicate that you removed all missing cases. E.g. the total for age group is 22434 but for BMI the total (N) is 22 435, and for education it is 22432. This is the case for all variables in the two time periods.

Line 158- correct the spelling for the word ‘overweight’

Line 163- The use of the word ‘commendable’ in the sentence is not appropriate, consider replacing it with ‘notable’. I am also not sure about ‘lower increase’ do you mean ‘small increase”???.

Please correct the title for table 2 accordingly. Thus according to what the table presents, not ‘row percentage’, e.g. Percentage of overweight/obese women by socioeconomic………………….

Under the subheading Risk factors of overweight and obesity and interaction results, the interpretation of results is not clear. It seems vague and incomprehensive. For instance, the following sentence needs revision; Among women of different age groups, the highest overweight and obesity risk were for the 2004/07 time period women in the age group 35-39 compared with women of 15-24 years of age.

Moreover, there are many typos in this section. The authors need to correct them. For instance in page 14, correct the following words, women for women, obesity for obesity, overweith for overweight asoocicated for associated…please correct all the typos.

In page 14, paragraph 2, the first sentence is not properly written, especially ‘associated with lessen the likelihood’………..

The last paragraph in the result section you present results on the interaction between wealth and TV viewing, wealth and education- but in the table these results are not presented.

Discussion

The first paragraph is a bit disorganised and needs some organization. One would assume that you would start by indicating the study aim as you have done, and not go much into elaborating the data source and the methodology since these are covered in the respective sections. In this paragraph give indicative findings on the prevalence of overweight/obesity among women and discuss those findings. There is also need to repeat results (in percentages) in this section, as a result use words such as increasing prevalence, in which socioeconomic groups etc. The discussion section generally needs to be strengthened. Findings should clearly be discussed in the light of the study objectives. As it is, it is very difficult to deduce that from the discussion. Much of the discussion is focused on prevalence rather than the interaction between wealth and place of residence. This section also needs to be revised for the correction of typos and grammatical errors.

References

Please make sure that all references cited in the article appear in the reference list and follow the PLOS One referencing style.

6. PLOS authors have the option to publish the peer review history of their article (what does this mean?). If published, this will include your full peer review and any attached files.

Reviewer #1: No

Reviewer #2: **Yes: **Dr Mpho Keetile, Department of Population Studies, University of Botswana, Gaborone, Botswana

---

## [Author Response · Author response to Decision Letter 0]

29 Oct 2020

Dear Editor,

Thank you for allowing us to revise our manuscript, now titled as “Interaction between the place of residence and wealth on the risk of overweight and obesity in Bangladeshi Women”. We have found the reviewers’ comments/feedback very helpful in improving the manuscript, and we have revised the manuscript accordingly. Additionally, we have addressed the journal requirements. Please find the revised manuscript along with this revised submission. 

Please note that, this research did not receive any specific grant from any funding agencies in public, commercial or not-for-profit sectors. Two authors (KA and TK) have affiliation from commercial organization, Purple Informatics (PI). The funder provided support in the form of salaries for TK, and consultancy fee for KA, but not for this paper work; and the funder did not have any additional role in the study design, data collection and analysis, decision to publish or preparation of the manuscript for this study. The specific roles of these authors are articulated in the ‘author contributions’ section on an attachment to this re-submission, however, we are re-stating in the following: 

Conceptualization: Kabir Ahmad

Data curation: Kabir Ahmad 

Formal analysis: Kabir Ahmad, Taslima Khanam 

Methodology: Kabir Ahmad, Taslima Khanam

Software: Kabir Ahmad 

Supervision: Enamul Kabir, Rasheda Khanam 

Writing – original draft: Kabir Ahmad, Taslima Khanam, Md. Irteja Islam, Syed Afroz Keramat

Writing – review & editing: Enamul Kabir, Rasheda Khanam

TK is an employee and KA is a consultant of the commercial affiliation, PI. These do not alter our adherence to PLOS ONE policies on sharing data and materials. Other authors do not have any competing interests.

The manuscript has not been submitted to or published in any other journal. Our point-by-point comments on the suggested revisions have been uploaded along with this re-submission in a file named as "response to reviewers". 

Best regards,

Kabir Ahmad (corresponding author)

PhD student, School of Commerce and Centre for Health Research 

University of Southern Queensland

Toowoomba, Queensland 4350, Australia

& 

Research Advisor, Purple Informatics,

Dhaka, Bangladesh

---

## [Decision Letter · Decision Letter 1]

20 Nov 2020

Interaction between the place of residence and wealth on the risk of overweight and obesity in Bangladeshi women

PONE-D-20-25454R1

Dear Authors,

We’re pleased to inform you that your manuscript has been judged scientifically suitable for publication and will be formally accepted for publication once it meets all outstanding technical requirements.

Kind regards,

Professor Hafiz T.A. Khan, Ph.D, CStat

Academic Editor

PLOS ONE

Additional Editor Comments (optional):

Reviewers' comments:

Reviewer's Responses to Questions

**Comments to the Author**

1. If the authors have adequately addressed your comments raised in a previous round of review and you feel that this manuscript is now acceptable for publication, you may indicate that here to bypass the “Comments to the Author” section, enter your conflict of interest statement in the “Confidential to Editor” section, and submit your "Accept" recommendation.

Reviewer #1: All comments have been addressed

Reviewer #2: All comments have been addressed

2. Is the manuscript technically sound, and do the data support the conclusions?

Reviewer #1: Yes

Reviewer #2: Yes

3. Has the statistical analysis been performed appropriately and rigorously? 

Reviewer #1: Yes

Reviewer #2: Yes

4. Have the authors made all data underlying the findings in their manuscript fully available?

Reviewer #1: Yes

Reviewer #2: Yes

5. Is the manuscript presented in an intelligible fashion and written in standard English?

Reviewer #1: Yes

Reviewer #2: Yes

6. Review Comments to the Author

Reviewer #1: (No Response)

Reviewer #2: The authors have addressed all the comments i had raised in the previous submission. I therefore think that the manuscript is ready for publication. Authors should make sure the references follow the journal style and all the cited references appear in the reference list.

7. PLOS authors have the option to publish the peer review history of their article (what does this mean?). If published, this will include your full peer review and any attached files.

Reviewer #1: No

Reviewer #2: **Yes: **Mpho Keetile, Ph.D. Department of Population Studies, University of Botswana.Gaborone. Botswana

---

## [Editor Report · Acceptance letter]

26 Nov 2020

PONE-D-20-25454R1 

Interaction between the place of residence and wealth on the risk of overweight and obesity in Bangladeshi women 

Dear Dr. Ahmad:

I'm pleased to inform you that your manuscript has been deemed suitable for publication in PLOS ONE. Congratulations! Your manuscript is now with our production department. 

Kind regards, 

on behalf of

Professor Hafiz T.A. Khan 

Academic Editor

PLOS ONE